# Study on Quantitative Characterization of Coupling Effect between Mining-Induced Coal-Rock Mass and Optical Fiber Sensing

**DOI:** 10.3390/s22135009

**Published:** 2022-07-02

**Authors:** Wengang Du, Jing Chai, Dingding Zhang, Yibo Ouyang, Yongliang Liu

**Affiliations:** 1College of Energy Engineering, Xi’an University of Science and Technology, Xi’an 710054, China; chaij@xust.edu.cn (J.C.); zhangdd@xust.edu.cn (D.Z.); 17629014025@163.com (Y.O.); ylliu.xust@stu.xust.edu.cn (Y.L.); 2Ministry of Education of the Western Mining and Mine Disaster Prevention and Control of Key Laboratory, Xi’an University of Science and Technology, Xi’an 710054, China; 3College of Safety Science and Engineering, Xi’an University of Science and Technology, Xi’an 710054, China

**Keywords:** mining rock mass, optical fiber sensing, deformation monitoring, coupling action

## Abstract

The monitoring of mine pressure, division of vertical zoning of the overburden, discrimination of key stratum structure of the overburden and monitoring of advanced abutment pressure are still the main research problems in the field of coal mining. Therefore, the promotion of development of a monitoring technology of mining-induced rock mass deformation has important research value in the mining field. There are many problems to be solved in the application of optical fiber sensing (OFS) to deformation monitoring, such as the corresponding relationship between actual deformation and optical parameters, the coupling relationship between the optical fiber and rock mass and the reasonable division of vertical zoning of the overburden. In this study, a quantitative index of coupling action between the mining rock mass and optical fiber is put forward, and the coupling coefficient of different vertical zonings is quantitatively analyzed and discussed. Based on this, five different media in contact with optical fiber are proposed. The relationship between the strain curve form, the development height of the fracture zone and the activity of key stratum is established. It is of great academic value and research significance to establish a characterization system of displacement, deformation and structural evolution of overlying strata based on optical fiber sensing technology.

## 1. Introduction

The crux to a scientific approach for controlling the safe and efficient production in a coal mine lies in the determination of the mining coal-rock mass motion behaviors [1,2,3]. Due to the lack of scientific understanding of deformation and motion laws of the surrounding rock caused by mining, roof fall, coal and gas outburst, mine water inrush, ground surface subsidence and other major catastrophic accidents and huge economic losses are often caused. Therefore, to promote the development of a monitoring technology of mining-induced rock mass deformation has important research value in the mining field [4,5,6]. Whether it is in situ rock mass deformation monitoring or laboratory model test research, the key point in researching the deformation law of mining rock mass is to obtain various deformation parameters scientifically and accurately [7,8,9,10].

From the concept of intelligent mining and sustainable development [11,12], research can promote the treatment of mine disasters and surface subsidence by studying the spatiotemporal evolution process of overburden movement, deformation monitoring of the rock mass and prediction of the vertical zoning height of the overburden. In view of the mining engineering problems, the stress field and deformation field are the most direct physical parameters for characterizing the development, migration and failure state of the rock fracture. With the continuous development of electronic information technology, there are more mature monitoring methods, such as the drilling video monitoring method [13], transient electromagnetic method [14], high-density resistivity method [15], acoustic CT tomography technology [16], geological radar [17], thermal infrared outfield and other advanced monitoring methods. All of the above methods belong to the multi-index evaluation system and cannot be used to characterize the overburden state with single test information. Displacement and strain of the surrounding rock mass are the most direct parameters for characterizing deformation. The traditional deformation measurement methods can only obtain superficial deformation information, which cannot realize the multi-scale distributed monitoring of the internal deformation of the rock mass structure. With the rapid development of the OFS technology, a new method has been provided for the deformation monitoring of mining-induced rock mass [18,19,20].

Scholars have conducted a lot of research works and achieved a series of results, among which a considerable number of research results have high engineering application value [21,22,23,24,25]. At present, the focus of research is to correctly understand the coupling relationship between optical fiber and rock mass in different deformation stages and to qualitatively or quantitatively analyze and characterize the coupling degree between optical fiber and rock mass. Scholars have conducted research on the above issues. Chai jing proposed the coupling criterion of optical fiber and rock mass based on the mechanical analysis of the response characteristics of optical fiber monitoring in the process of mining rock mass deformation and established the optical fiber monitoring model of rock mass deformation [26]. She junkuan believes that the deformation coordination between the embedded optical fiber sensing and the rock and soil mass has a significant impact on the monitoring results. With the help of the laboratory pull-out test, the pull-out force pull-out displacement curves of the optical fiber and the sand and soil mass under three different loads are compared [27]. Li Bo designed a three-point bending test of the soil strip, buried the distributed optical fiber in the soil and analyzed the coupling performance of the optical fiber and the soil in combination with the PIV photogrammetry technology [28]. Wuhan carried out a feasibility study on soil shear displacement monitoring by using OFDR technology and particle image velocimetry (PIV). The study found that the calculated value of optical cable elongation and soil shear displacement showed a linear relationship at the stage of good coupling between the optical cable and soil. Setting anchor points on the optical fiber or improving soil confining pressure can increase the coupling between the optical cable and rock mass, so as to obtain a better monitoring effect [29]. There are few reports on the analysis of the coupling relationship between optical fiber and the surrounding rock mass in the process of large-scale high-strength deformation of overlying rock mass caused by underground coal mining.

At present, the application of the optical fiber sensing technology in mining engineering is still in the exploratory stage. There are many problems to be solved in the application of the optical fiber sensing technology to the deformation monitoring of mining rock mass, such as the reliability verification of deformation information detection, the corresponding relationship between actual deformation of the rock mass and the detection optical parameters, coupling relationship between the optical fiber and rock mass and the reasonable characterization of division of vertical zoning in the overburden and weighting events in stopes. How to obtain the deformation information and stress evolution law of the mining overburden scientifically and effectively through optical fiber sensing technology has become the focus of current research. Based on this, in view of the insufficient analysis of the coupling relationship between the optical fiber and mining rock mass in different mining stages, a quantitative index of the coupling relationship between optical fiber sensing and mining rock mass is proposed in this paper, and the influence of spatiotemporal evolution process of two dimensional variables, i.e., vertical layer height and horizontal working face advancing position, on the coupling coefficient is discussed. It is of great academic value and research significance to establish a characterization system of displacement, deformation and structural evolution of overlying strata based on optical fiber sensing technology.

## 2. Structural Characteristics and Evolution Law of Mining Overburden

### 2.1. Evolution Process of Overburden in Different Mining Stages

During the period from the opening off-cut to the initial fracture of the basic roof, due to the small mining space, the basic roof has a certain degree of bending and sinking movement, which can be regarded as a fixed beam structure at both ends. When the overhanging length of the rock beam exceeds the strength limit it can bear, it will crack and form the first weighting event. At this time, the broken rock beam forms a cantilever beam structure with one end fixed and one end overhanging, and the stable rock mass above it forms a new fixed beam structure. As the working face continues to advance, the cantilever beam structure reaches its limit-breaking span, which causes periodic instability and fracture of the basic roof, causing periodic weighting phenomena. The structure evolution process of the overburden rock is shown in Figure 1.

As shown in Figure 2, in the initial mining stage, the basic roof forms a two-end fixed beam structure, the front and rear coal walls are beam supports, and the load of overlying strata is equivalent to the uniformly distributed load “*q*”, which can be calculated according to the following equation:(1)$$q=\gamma_{1}h_{1}+\gamma_{2}h_{2}+\gamma_{3}h_{3}+\ldots \gamma_{n}h_{n}$$
where *γ* is the bulk density of rock stratum, “*h*” is the thickness of rock stratum, “1, 2, 3…*n*” is the sequence number of rock stratum. According to the mechanics theory, the reaction forces “*Fa*, *Fb*” and bending moment “*M*” are formed at the fixed ends of the rock beam; the expression is
(2)$$F_{a}=F_{b}=\frac{1}{2}ql$$

The shear force and bending moment at any section *x* away from the left end are as follows:(3)$$\begin{array}{l} F_{x}=F_{a}-qx=\frac{1}{2}q(l-2x)\\ M_{x}=F_{a}-\frac{1}{2}qx^{2}-M=\frac{q}{12}(6lx-6x^{2}-l^{2}) \end{array}$$

The analysis shows that the maximum shear force and bending moment of the beam are at both ends:(4)$$\begin{array}{l} F_{\max}=\pm \frac{1}{2}ql\\ M_{\max}=-\frac{1}{12}ql^{2} \end{array}$$

From the bending moment distribution diagram of the beam, it can be seen that the upper part of the neutral plane at both ends of the beam is compressed, the middle part of the beam and the lower part of the neutral plane are under tension, and the tensile stress concentration at the middle point is the highest. The compressive strength of the rock material is far greater than the tensile strength, so it is easy to produce tensile failure.

Therefore, with the continuous expansion of the mining space, when the overhanging length of the beam is greater than the ultimate load it can bear, the tensile failure will occur in the middle of the lower surface and fracture. The tensile strength [σt] of the rock mass can be obtained with a rock mechanics test, and the maximum tensile stress “*σ*_max_” of the rock beam structure is as follows:(5)$$\sigma_{\max}=\frac{M_{\max}}{W_{z}}$$
where *W_Z_* is the bending section coefficient of the beam structure, which can be obtained from the thickness and height of the beam
(6)$$W_{Z}=\frac{bh^{2}}{6}$$
where *b* is the thickness of the beam, and h is the width of the beam. When *σ*_max_ > [*σ*_t_], the basic roof rock beam structure will break for the first time and form the initial weighting event. Thus, the first weighting step length is obtained as follows:(7)$$L_{ini}=h\sqrt{\frac{2\left[\sigma_{t}\right]\cdot b}{q}}$$

The 52,304 working face of Daliuta coal mine in Shendong mining area, China, is taken as the geological prototype in this paper. The thickness of the coal seam in the 52,304 working face is 6.6~7.3 m. The structure of the coal seam is simple, and the dip angle is 1~3°. The basic roof is siltstone with a thickness of 5.2 to 28.3 m. The average uniaxial compressive strength of the basic roof is 30.57 MPa, and the average tensile strength is 2.28 MPa. Substituting the parameters, the first weighting step length of the 52,304 working face is 79.34 m.

### 2.2. Vertical Zoning Theory of Mining Rock Mass and Optical Fiber Sensing Mechanism

After the coal seam is mined out, the overlying strata collapse from the bottom to the top, and the farther away they are from the working face, the less affected they are by mining. The deformation degree of the rock mass at different heights is obviously different, forming different independent zoning areas. The classic “three zones (caved zone, fractured zone and bending subsidence zone)” theory was first put forward by Soviet scholars in 1958 [30].

As shown in Figure 3, when the fiber is in the mined-out area, the stress state in different areas is significantly different. The red part is located in the bending subsidence zone. The typical characteristics of this area are that the rock mass structure is complete, macro cracks are not developed, and the coupling contact relationship between the fiber and rock mass is positive. The green part is the separation zone formed under the key stratum. The optical fiber is separated from the rock mass in the area, and the measured data cannot reflect the real deformation of the rock mass. The yellow area is in the fracture zone and caved zone, the rock mass stress reaches the strength limit and then fractures, and there is friction and squeezing force between the broken rock block and the optical fiber. Therefore, the vertical zoning characteristics of the overburden can be inversed by analyzing the coupling contact relationship between the optical fiber and rock mass in different zoning areas.

Field detection is the most direct and effective method for the development of height of the water-conducted fracture zone. Generally, the height can be obtained indirectly by measuring the leakage of the drilling fluid, as shown in Table 1. The combination of basic mining problems and optical fiber sensing technology is of great significance for promoting the application and development of optical fiber sensing technology in the mining engineering field.

### 2.3. Development Law of Roof Crack in Coal Mining

The original meaning of “weighting events” refers to the process in which the working resistance of the supports in the working face is significantly increased due to the roof strata movement, so it is the most direct and effective research method to analyze weighting events in the stope by analyzing the working resistance of the supports in the field. According to the analysis of roof pressure caused by coal seam mining, the deflection and bending moment of key stratum are important indices for predicting its fracture.

As shown in Figure 4a, before the initial weighting event of the roof, all the rock formations under the first key stratum can be regarded as the equivalent direct roof, which will gradually collapse with the expansion of the mining space. When the overhanging length of the lower key stratum reaches a certain value, bending deformation will occur. Whether the rotary space in the lower part is sufficient will directly determine the initial weighting strength. At this time, the mining energy has not spread to the upper key stratum. In this case, the lower key stratum can be simplified as a fixed beam structure at both ends. The differential equation of the deflection curve is as follows:(8)$$\frac{d^{2}\omega }{dx^{2}}=\frac{M}{EI}$$

The above equation is the differential equation of the deflection curve, and the deflection curve equation can be obtained by integrating the above equation as follows:(9)$$\omega =\frac{qx}{24EI}(l^{3}-2lx^{2}+x^{3})$$
where *q* is the uniformly distributed load on the lower key stratum, *E* is the elastic modulus of the key stratum rock mass, *I* is the inertia moment of the rectangular section, l is the suspended length of the rock beam. The maximum deflection is located in the middle of the rock beam:(10)$$\omega_{\max}=\frac{5ql^{4\zeta }}{384EI}=\frac{5\zeta ql^{4\zeta }}{32EH_{L}}$$
where *ς* is the deflection correction coefficient; *H_L_* is the thickness of the lower key layer. Taking the maximum allowable subsidence value of the rock beam in the lower key stratum as the judging condition for the first weighting of the roof, then
(11)$$\left[w\right]=h_{c}-\omega_{\max}=m-\sum_{1}^{i}h_{i}(K_{S}-1)-\frac{5ql^{4\zeta }}{32EH_{L}{}^{3}}$$
where [*w*] is the maximum allowable subsidence of the lower key stratum; *m* is the mining height; Σ*h_i_* is the thickness of the equivalent direct roof; *K*_S_ is the crushing expansion coefficient. When [*w*] > 0, it indicates that the fault rock blocks formed by the equivalent direct roof collapse are not in contact with the lower key stratum, and the space formed can be used for rotation and sinking of the key stratum. The larger the [*w*], the stronger the initial weighting strength; on the contrary, when [*w*] ≤ 0, it indicates that the fault rock blocks formed by the equivalent direct roof are connected to the top, and the key stratum does not have enough space to rotate and sink to form the dynamic load. The ground pressure phenomenon in this process is milder than the former. Through calculation, the maximum allowable subsidence of the lower key stratum is 0.3 m, which shows that there is a relatively rich rotary subsidence space under the key stratum after the instability and fracture, so the weighting strength is intense.

## 3. Basic Theory of Optical Fiber Sensor for Rock Mass Deformation Monitoring

### 3.1. Principle of Distributed Optical Fiber Testing (BOTDA)

When the ambient temperature changes or the optical fiber deforms, the sound velocity and the refractive index of light in the fiber will change, which will change the Brillouin frequency shift in the optical fiber, as shown in Figure 5. The Brillouin frequency shift is linear with the strain and temperature of the optical fiber.

Without considering the influence of temperature, the relationship between the Brillouin frequency shift and strain can be expressed as follows:(12)$$\nu_{B}(\varepsilon ,T_{0})=\frac{2\nu_{p}}{C}n(\varepsilon ,T_{0})\sqrt{\frac{\left[1-k(\varepsilon ,T_{0})\right]E(\varepsilon ,T_{0})}{\left[1+k(\varepsilon ,T_{0})\right]\left[1-2k(\varepsilon ,T_{0})\right]\rho (\varepsilon ,T_{0})}}$$

The optical fiber core material is generally silicon dioxide, which has a large elastic modulus and belongs to the brittle material, so the tensile strain is smaller than that of the general materials. In the practical application of structural health monitoring, the small deformation stage before structural failure is mainly detected. The Taylor expansion of the variables in Equation (12), which are related to the strain at the zero point, is as follows:(13)$$\left\{\begin{array}{l} n(\varepsilon ,T_{0})=n(0,T_{0})+{\left(\frac{dn(\varepsilon ,T_{0})}{d\varepsilon }\right)}_{\varepsilon =0}\cdot \varepsilon \\ E(\varepsilon ,T_{0})=E(0,T_{0})+{\left(\frac{dE(\varepsilon ,T_{0})}{d\varepsilon }\right)}_{\varepsilon =0}\cdot \varepsilon \\ k(\varepsilon ,T_{0})=k(0,T_{0})+{\left(\frac{dk(\varepsilon ,T_{0})}{d\varepsilon }\right)}_{\varepsilon =0}\cdot \varepsilon \\ \rho (\varepsilon ,T_{0})=\rho (0,T_{0})+{\left(\frac{d\rho (\varepsilon ,T_{0})}{d\varepsilon }\right)}_{\varepsilon =0}\cdot \varepsilon \end{array}\right.$$

By substituting Equation (13) into Equation (12) and expanding the quadratic term, the relationship can be obtained by taking the first-order term of *ε*:(14)$$\nu_{B}(\varepsilon ,T_{0})=\nu_{B}(0,T_{0})\left\{\begin{array}{l} 1+\frac{n_{\varepsilon }{}^{\prime }}{n(0,T_{0})}\varepsilon +\frac{E_{\varepsilon }{}^{\prime }}{2E(0,T_{0})}\varepsilon -\frac{\rho_{\varepsilon }{}^{\prime }}{2\rho (0,T_{0})}\varepsilon \\ +\frac{k_{\varepsilon }{}^{\prime }k(0,T_{0})[2-k(0,T_{0})]}{\left[1-k^{2}(0,T_{0})][1-2k(0,T_{0})\right]}\varepsilon \end{array}\right\}$$

In order to simplify the expression of the equation, let
(15)$$\left\{\begin{array}{l} \Delta n_{\varepsilon }=\frac{n_{\varepsilon }{}^{\prime }}{n(0,T_{0})}\\ \Delta E_{\varepsilon }=\frac{E_{\varepsilon }{}^{\prime }}{2E(0,T_{0})}\\ \Delta \rho_{\varepsilon }=\frac{-\rho_{\varepsilon }{}^{\prime }}{2\rho (0,T_{0})}\\ \Delta k_{\varepsilon }=\frac{k_{\varepsilon }{}^{\prime }k(0,T_{0})[2-k(0,T_{0})]}{\left[1-k^{2}(0,T_{0})][1-2k(0,T_{0})\right]} \end{array}\right.$$

After simplification, Equation (14) can be rewritten as follows:(16)$$\nu_{B}(\varepsilon ,T_{0})=\nu_{B}(0,T_{0})\left[1+(\Delta n_{\varepsilon }+\Delta \rho_{\varepsilon }+\Delta E_{\varepsilon }+\Delta k_{\varepsilon })\varepsilon \right]$$

It can be seen from the above equation that four constant coefficients (Δ*n**ε*, Δ*ρε*, Δ*E**ε*, Δ*k**ε*) should be determined in order to obtain the relationship between the Brillouin frequency shift and strain without considering the influence of temperature. Ordinary quartz optical fiber at normal atmospheric temperature: Δ*n**ε* = −0.22; Δ*ρε* = 0.33; Δ*k**ε* = 1.49; Δ*E**ε* = 2.88. The relationship between the Brillouin frequency shift and strain and temperature can be expressed as follows:(17)$$\nu_{B}(\varepsilon ,T)=\nu_{B}(0)+\frac{d\nu_{B}(\varepsilon )}{d\varepsilon }\varepsilon +\frac{d\nu_{B}(T)}{dT}(T-T_{0})$$
where *ν_B_*(*ε*, *T*) is the Brillouin frequency shift; *dv_B_*(*T*)/*dt* is the temperature sensitivity coefficient; *ν_B_* (*ε*) is the strain-induced Brillouin shift; *dv_B_*(*ε*)/*dε* is the deformation sensitivity coefficient; *ν_B_*(0) is the initial Brillouin frequency shift.

### 3.2. Coupling Relationship between Mining Rock Mass and Optical Fiber

#### 3.2.1. Mechanical Behavior of Rock–Optic Fiber Interface

In practical application, it is found that the deformation coordination of optical fiber sensing and the rock structure has a direct impact on the monitoring results. When a similar material is consolidated, there is no gap between the fiber and the rock mass occurring before large deformation. In this stage, the strain transmission rate between optical fiber and the surrounding rock is the highest. In addition to the absorption of elastic deformation energy by optical fiber cladding and sheath, the strain measured by optical fiber is the strain of the rock mass itself. Generally, the corresponding region is where the working face closes to the optical fiber buried position, and the surrounding rock structure is where the located optical fiber is complete. When the working face passes through the optical fiber, the fiber is in the mined-out area, and the surrounding rock is where the located optical fiber starts to be affected by the mining, and the movement of sinking, separation, breaking and caving occurs continuously. At this time, the stress state of the optical fiber is extremely complex. The deformation scale of the mining overburden weakens in turn from the bottom to the top, with different zoning characteristics, and the optical fiber has different data response characteristics in different vertical zoning ranges.

The process of rock deformation, the fiber core, coating and sheath are subject to different degrees of tension. In the process of progressive failure of the fiber–rock interface, the mechanical behavior of the interface can be characterized by ideal elastic-plastic model. Zhang et al. proposed the fiber–soil interface progressive failure pull-out model [31]. The model is based on the ideal elastic-plastic model of the interfacial shear stress–strain. The pull-out failure process of the optical fiber in soil is divided into three stages: pure elastic, elastic-plastic and pure plastic. As shown in Figure 6, the relationship between the pull-out force P and displacement μ_0_ is as follows:(18)$$P=\left\{\begin{array}{l} -\frac{2DG}{\beta }\tanh(\beta L)\mu_{0}\;\;\;\;\;\;\;\;\;(\mathrm{pure}\;\mathrm{elastic}\;\mathrm{stage})\\ -\frac{AE}{L_{P}}(\mu_{0}+\frac{\tau_{\max}}{G})+\frac{\pi D}{2}L_{p}\tau_{\max}\;\;\;\;(\mathrm{elastic}\text{-}\mathrm{plastic}\;\mathrm{stage})\\ \pi D\cdot \tau_{\max}L\;\;\;\;\;\;\;\;\;\;(\mathrm{pure}\;\mathrm{plastic}\;\mathrm{stage})\\ \beta =\sqrt{4G/ED} \end{array}\right.$$
where *D*, *l*, *E* and *A* are the diameter, embedded length, elastic modulus and cross-sectional area of the optical cable, respectively; *G* is the shear coefficient of the fiber–rock interface; *τ*_Max_ is the shear strength of the interface; *L_P_* is the plastic failure length in the process of interface failure. The corresponding strain expression of each stage is as follows:(19)$$\varepsilon (x)=\left\{\begin{array}{l} \frac{P}{AE}\cdot \frac{\sinh\beta (L-x)}{\sinh\beta L}\;\;\;\;\;\;\;\;\;(\mathrm{pure}\;\mathrm{elastic}\;\mathrm{stage})\\ \frac{F_{T}}{AE}\cdot \frac{\sinh\beta (L-x)}{\sinh\beta (L-L_{P})}\;\;\;\;\;\;\;\;\;\;(\mathrm{elastic}\;\mathrm{part}\;\mathrm{of}\;\mathrm{elastic}\text{-}\mathrm{plastic}\;\mathrm{stage})\\ \frac{4\tau_{\max}}{DE}\cdot (L_{P}-x)+\frac{4F_{T}}{\pi DE}\;\;\;\;\;\;(\mathrm{plastic}\;\mathrm{part}\;\mathrm{of}\;\mathrm{elastic}\text{-}\mathrm{plastic}\;\mathrm{stage})\\ \frac{4\tau_{\max}}{DE}\cdot (L-x)\;\;\;\;\;\;\;\;\;\;(\mathrm{pure}\;\mathrm{elastic}\;\mathrm{stage}) \end{array}\right.$$

In the equation, the value range of × is from 0 to l; the value range of the elastic part is from *l*_p_ to l, and that of the plastic part is from 0 to *l*_p_; *F*_T_ is the axial force at the transition point from the elastic to plastic stage. In this study, the diameter of the optical fiber used is 0.002 m, buried length is 1.2 m, cross section area is 3.14 × 1016 m^2^, elastic modulus is 0.37 GPa. After calculation, it can be concluded that the shear strength of the rock–fiber interface is 7.2 kPa, which increases with the increase in confining pressure. When the shear stress between the fiber and the rock mass exceeds 7.2 kPa, the fiber will slide from the rock mass.

#### 3.2.2. Quantitative Analysis of Rock–Fiber Coupling Relationship

##### Theoretical Model of Quantitative Analysis of Coupling Relationship

The coupling effect of the optical fiber and the rock mass can be divided into three stages. In the first stage, there is no discontinuous deformation in the surrounding rock where the optical fiber is located. The rock mass structure is complete and in close contact with the optical fiber. In the second stage, when the surrounding rock begins to produce discontinuous deformation, such as cracks and separation, the shear layer of the rock interface gradually incurs shear failure and interface peeling. In the third stage, when the rock mass is unstable and cracked, the interface between the optical fiber and the rock mass will slide relatively. The downward movement and compaction of the unstable rock block will cause the optical fiber tensile stress concentration. Figure 7 shows the theoretical relationship model between the coupling degree of the optical fiber–rock mass and the deformation degree of the rock mass.

##### Mathematical Expression of Coupling Coefficient between Optical Fiber and Rock Mass

In view of the application of optical fiber sensing technology to the deformation monitoring of mining rock mass, due to the large deformation scale of the overlying strata after coal seam mining, there is a great difference in deformation scale from micro deformation to macro deformation. It is defined that the coupling coefficient varies in the range of 0~1. When the coefficient is 1, the coupling degree is the highest, and when the coefficient is 0, the rock mass is completely separated from the fiber. According to the corresponding relationship between the multiple model tests phenomena and the coupling coefficient, the gradation of the coupling coefficient is determined, as shown in Table 2.

The strain detected by the distributed optical fiber sensors at different height positions also has the zoning characteristic in the vertical direction, and the development trend of the strain curve is similar in a certain range of the height, and it will change significantly when it breaks through this height. By calculating the relative error between the strain at a different height and the mining position and the average strain in the whole excavation process, and then homogenizing, the coupling contact degree between the optical fiber and the rock mass is analyzed with optical fiber strain sensing, which provides a theoretical basis for reasonable determination of the overlying rock zoning and stage division. The coupling coefficient of the optical fiber and the rock mass can be expressed as follows:(20)$$k_{j}=\left|\frac{\left|f(x)-\frac{1}{n}\sum_{i=1}^{n}\varepsilon_{i}\right|}{\frac{1}{n}\sum_{i=1}^{n}\varepsilon_{i}}-1\right|$$
where *k*_j_ is the coupling coefficient of the optical fiber and the rock mass in the *j*-th excavation; *x* is the advancing distance of the *j*-th excavation; *f*(*x*) is the strain value of the corresponding advancing position; *n* is the total number of excavations; and *ε_i_* is the strain of the arbitrary sequence excavation.

##### Normalization of Coupling Coefficient

The coupling coefficient is defined as a constant varying from 0 to 1, and the normalization of the coupling coefficient obtained from Equation (20) is the premise of quantitative analysis. Normalization consists of limiting the data that need to be processed (through some algorithm) to a specific range of needs. Its specific function is to sum up the statistical distribution of the unified samples. The linear method is selected for the normalization of the optical fiber–rock mass coupling coefficient in this paper. The expressions of normalization and standardization are [32]:(21)$$\begin{array}{l} \mathrm{normalization}:\;\frac{x_{i}-x_{\min}}{x_{\max}-x_{\min}}\\ \mathrm{standardization}:\;\frac{x_{i}-\mu }{\sigma } \end{array}$$
where *μ* and *σ* represent the mean and standard deviation of the sample, *x*_max_ is the maximum value of the sample, and *x*_min_ is the minimum value of the sample. Normalization and standardization are both linear transformations, in essence. Under the premise of the given data, set the constant
(22)$$\begin{array}{l} \alpha =x_{\max}-x_{\min}\\ \beta =x_{\min} \end{array}$$

From the above equation, it can be seen that normalization is the same as standardization. The above equation is deformed:(23)$$\left\{\begin{array}{l} \mathrm{normalization}:\;\frac{x_{i}-\beta }{\alpha }=\frac{x_{i}}{\alpha }-\frac{\beta }{\alpha }=\frac{x_{i}}{\alpha }-C\\ \mathrm{standardization}:\;\frac{x_{i}-\mu }{\sigma }=\frac{x_{i}}{\sigma }-\frac{\mu }{\sigma }=\frac{x_{i}}{\sigma }-D \end{array}\right.$$

In the equation, *C* and *D* are constants. It can be seen that both normalization and standardization are linear transformations in which the sample data are first scaled and then translated. From the data output range, the normalized output range is between 0 and 1, while the standardized output range is between negative infinity and positive infinity.

### 3.3. Establishment of Physical Similarity Model Based on Optical Fiber Sensing

Considering the simulated rock formation thickness and the geometric size of the laboratory model frame, the geometric size of the model is 1500 (length) × 600 (width) × 1300 (height) mm. The geometric similarity ratio is 1:150, the thickness of the overlying strata is 1.28 m when the thickness of the model coal seam is 5 cm, which meets the height limit of the model frame, and it can be simulated to the ground surface loose layer without external load replenishment [33]. According to the requirements of similarity and previous research experience, it is determined that sand, pulverized coal, clay and mica (also as layered materials) are used as aggregates in each rock layer. Gypsum and lime powder are used as cementitious material and water as solvent material. The measurement methods used in this physical similarity model test include distributed optical fiber strain measurement system (BOTDA) and FBG. A total of four vertical optical fibers are arranged, named V1\V2\V3\V4, as shown in Figure 8. The sensing optical fiber needs to determine the specific position coordinates of each optical fiber in the model through spatial positioning [34]. The distributed optical fiber monitoring demodulation equipment is “NBX-6055”.

## 4. Characterization of Vertical Zoning Division of Mining Overburden Based on Analysis of Coupling Relationship between Rock Mass and Optical Fiber

### 4.1. Vertical Zoning Characteristic of Monitoring Strain Data

In order to obtain the variation law of the strain in a different height, the data collected from the physical model test are analyzed. The total height of the model is 129 cm. Starting from the bottom plate, the monitoring points are taken from the bottom to the top of the model at an interval of 10 cm. The strain distribution of each measuring point with the advance of the working face is shown in Figure 9. The data of strain curves with similar variation trend are put in the same graph, and the dotted line is used as the reference for the obvious change of strain, which shows that the trend of the strain curve will change significantly when breaking through a certain height.

It can be found that the strain development trend is basically the same from the height of 1–30 cm, the negative strain increases with the advancing of the working face from 0 to 45 cm, and the strain increases rapidly from negative to positive when the working face advances to 45 cm, where the working face passes through the fiber. The failure and instability of the sub key stratum when the working face advances to 66 cm promotes the strain to increase in the second round and then to decrease and stabilize.

The results show that the strain curve in the height of 40–50 cm has a great change compared with that before; in particular, the strain value decreases substantially after the working face advances to 60 cm. The development trend of the curve in the range of 60–80 cm height is basically the same; compared with the height of 50 cm, the position where the strain increases significantly from zero is changed from the working face advancing to 45–60 cm. The development trend of the curve in the range of 90–110 cm height is basically the same; the position where the strain increases significantly from zero is changed from the working face advancing to 66–90 cm. The development trend of the curve in the range of 120–129 cm height is basically the same, the strain value is far less than that under it, and the strain at the top of the model is almost zero.

### 4.2. Coupling Model of Optical Fiber–Rock Mass

In order to obtain the coupling function relationship between the optical fiber and the rock mass in the vertical zoning area, the strain curves of the representative height position in each zone are fitted. Starting from the top of the model, the strain curves at the height of 120, 100, 70, 50 and 20 cm are taken for nonlinear fitting analysis. The Gauss function is widely used in natural science, and the figure is like a clock hanging upside down. The fitting results are shown in Figure 10.

The strain of the rock mass changes dynamically with the advancement of the working face. Similarly, the coupling coefficient between the rock mass and the optical fiber is also dynamic. Therefore, in order to realize the quantitative characterization of the coupling relationship between the optical fiber and the rock mass, it is necessary to control the variable in the horizontal or vertical dimension. Monitoring data indicate that the development of the strain curve is mainly controlled by two factors: in the horizontal direction, it is affected by the relative position of the optical fiber and the working face; in the vertical direction, it is affected by the instability and fracture of the key stratum.

The strain curves of different heights in the vertical direction can be divided into five zones according to the variation trend. It is precisely because of the great difference in the coupling relationship between the optical fiber and the rock mass in the five different regions that the strain difference in optical fiber detection in different regions is caused. From the top to the bottom, 120–129 cm is weakly affected by mining, and the rock mass structure is complete, which is the upper fixed end of the optical fiber; the rock mass in the height of 90–110 cm above the main key stratum is protected by the main key stratum; the deformation is small at the initial stage of mining; when the main key stratum becomes unstable, the strain increases sharply. In the height range of 60–80 cm, the rock mass under the main key stratum and above the sub key stratum incurs strain surge after the sub key stratum becomes unstable. The height range of 40–50 cm is the separation empty area formed above the caving zone, and the optical fiber does not contact any substance. The height range of 1–30 cm is the damage area and the coal seam floor.

For the mining-affected area, the strain curve shows a trend of increasing first, then decreasing, then stabilizing and then increasing again. The position of the first strain increase is the instability position of the sub key stratum, and the position of the second strain increase is the instability position of the main key stratum. According to the fitting results, the coupling coefficient of fiber–rock can be expressed as [32]:(24)$$k_{j}=\left|\frac{\left|f(x)-\frac{1}{n}\sum_{i=1}^{n}\varepsilon_{i}\right|}{\frac{1}{n}\sum_{i=1}^{n}\varepsilon_{i}}-1\right|=\left|\frac{\left|y_{0}+(\frac{A}{w\times \sqrt{\pi /2}})\cdot e^{-2{\left[\frac{\left(x-x_{c}\right)}{w}\right]}^{2}}-\frac{1}{n}\sum_{i=1}^{n}\varepsilon_{i}\right|}{\frac{1}{n}\sum_{i=1}^{n}\varepsilon_{i}}-1\right|$$
where *k_j_* is the coupling coefficient of fiber–rock in the *j*-th excavation; *x* is the advancing distance of the working face in the *j*-th excavation; *f*(*x*) is the strain value of the corresponding advancing position; n is the total number of excavations; and *ε_i_* is the strain of the arbitrary sequence excavation. *y*_0_, *xc*, *w* and *A* are the four known constants in the Gaussian function obtained from the above fitting results. According to Equation (24), the coupling coefficients of fiber–rock are calculated at the height of 120, 100, 70, 50 and 20 cm when the working face advances to the positions of 21, 45, 66, 84, 90 and 111 cm, respectively. As shown in Table 3.

At the height of 120 cm, the coupling coefficient of the whole advancing process is always kept at 1, which indicates that the area is almost unaffected by mining, the rock structure is intact and keeps close contact with optical fiber, and the strain measured by the optical fiber is the real strain of the rock mass. The rock mass in the height of 100 cm is above the main key stratum, and the coupling coefficient continually decreases slowly under the protection of the main key stratum. Through the above-mentioned interface mechanical analysis, it can be seen that the coupling relationship of fiber–rock is always synchronous, coupled and compatible. The coupling coefficient suddenly drops to less than 0.5 at the height of 70 cm and under it when the working face advances to 66 cm. From the above analysis, it can be seen that the first instability and fracture occur in the sub key stratum. At the height of 50 cm, the rock mass is completely separated from the optical fiber in the large abscission layer empty space, and the coupling coefficient is 0. The mined-out area is gradually compacted as the working face continues to advance, and the coupling coefficient picks up gradually.

From Figure 11, the whole process of the working face advance can be divided into four typical characteristic stages. The coupling coefficient decreases obviously for the first time when the working face passes through the optical fiber; the second significant change point is that after the first instability of the sub key stratum. The coupling coefficient decreases to 0–0.4, the fracture of the sub key stratum causes the large-scale movement and failure of the overlying strata, and the optical fiber separates from the rock mass in some areas. The third significant change point occurs after the first instability of the main key stratum. It can be seen that the coupling coefficient of 70 cm height decreases from 0.6 to 0.4, and the coupling coefficient of the lower failure zone increases obviously. This is mainly due to the failure and instability of the key stratum, which causes the strong compaction of the damaged area in the mined-out area, which leads to the increase in the coupling coefficient in the lower part of the mined-out area. The results show that the response of the optical fiber sensing of rock mass deformation is mainly affected by the relative position of the working face and optical fiber and the instability of the key stratum.

### 4.3. Characterization of Vertical Zoning of Overburden Based on Optical Fiber Sensing

To sum up, the vertical zoning of overlying strata can be scientifically divided with the coupling coefficient of the optical fiber and the rock mass. The coupling coefficient of the whole working face advancing process is calculated as shown in Figure 12. The evolution process of the contact relationship between the optical fiber and the surrounding rock can be clearly reflected by it. The main mutation position is the instability point of the key stratums and the position where the working face passes through the fiber. From the first instability of the first key stratum to the first instability of the second key stratum, the coupling coefficient between the model height of 40 and 50 cm is reduced to around 0. This area is a semi-elliptical space formed above the collapsed gangue in the mined-out area, and the optical fiber is completely separated from the rock mass. When the rock mass of the higher layer is unstable and collapsed, the semi-elliptical space is gradually transferred to the upper position. Because the rock mass is broken and expanded, the separation gap is gradually filled, and the fiber–rock coupling coefficient rises.

Finally, the water-conducted fractured zone develops in the lower part of the main key stratum and stops. Longitudinal cracks are developed on both sides of the main key stratum in the model test. The rock mass structure above the whole main key stratum is complete, and no cracks occur. It can be seen from the figure that the cloud picture of the coupling coefficient above the height of 95 cm is orange-red, and that below 95 cm height is green-blue-purple. Taking the green area and the orange area in the cloud picture as the boundary, the coupling coefficient of 0.65 is taken as the boundary between the fractured zone and the unbroken area of the mining overburden, that is, the upper limit of the development height of the water-conducted fractured zone.

The figure on the right in Figure 12 shows the spatiotemporal evolution process of mining rock mass deformation obtained by distributed optical fiber monitoring. According to the advancing direction of the working face, the strain distribution forms five steps, and the step height increases gradually. The first stage is the process of advancing 0–45 cm, and the corresponding working face is close to the optical fiber; the second stage is the process of advancing 45–66 cm, and the corresponding working face passes through the optical fiber to the instability of the first key stratum; the third stage is the process of advancing 66–90 cm, corresponding to the process between the instability of the first key stratum and the second key stratum; the fourth stage is the process of advancing 90–111 cm, corresponding to the instability of the second key stratum to the third key stratum; the fifth stage is the process from the instability of the third key stratum to the end of mining. The first instability of the three key strata causes the overburden to move violently, and the strain mutation is obvious.

According to the theoretical analysis, the overlying strata in the 52,304 working face of the Daliuta coal mine belong to the category of hard roof conditions, and the height of the water-conducted fractured zone can be conservatively taken as 18~28 times of the mining height. According to the theoretical calculation, the development height of the water-conducted fractured zone is 128.52–199.92 m. The height of the water-conducted fractured zone detected by optical fiber sensing in the model test is 142.5 m (95 × 150 cm). With the help of optical fiber sensing monitoring, the development height of the water-conducted fractured zone can be accurately predicted.

## 5. Conclusions

(1)The coupling effect of the fiber–rock can be divided into three stages. In the first stage, there is no discontinuous deformation in the surrounding rock where the optical fiber is located. In the second stage, when the surrounding rock begins to produce discontinuous deformation, the shear layer of the rock interface gradually incurs shear failure and interface peeling. In the third stage, when the rock mass is unstable and cracked, the interface between the optical fiber and the rock mass will slide relatively.(2)The strain curves of different heights in the vertical direction can be divided into five zones according to the variation trend. It is because of the great difference in the coupling relationship between the fiber and the rock in the five different regions that the strain difference of optical fiber detection in different regions is caused. Area 1 is the original rock stress zone; area 2 is the bending subsidence zone; the sum of area 3 and 4 is the fractured zone; area 5 is the coal seam floor.(3)A quantitative index of coupling between the mining rock mass and the fiber is proposed: the coupling coefficient. The coupling coefficient of 0.65 is taken as the boundary between the fractured zone and the unbroken area of the mining overburden, that is, the upper limit of the development height of the water-conducted fractured zone.

## Figures and Tables

**Figure 1 sensors-22-05009-f001:**
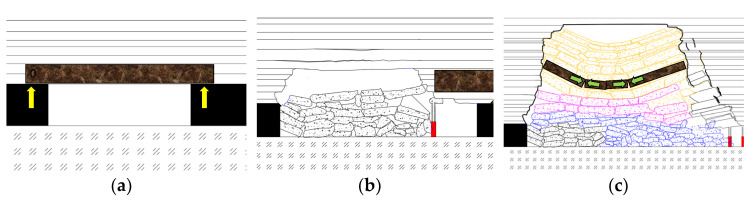
Structure evolution process of mining overburden. (**a**) Fixed beam structure; (**b**) cantilever beam structure; (**c**) masonry beam structure.

**Figure 2 sensors-22-05009-f002:**
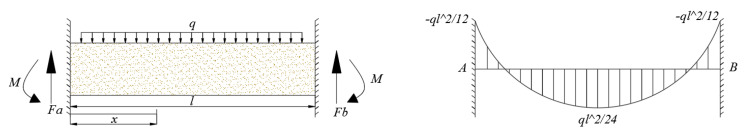
Mechanical model of roof fixed beam in initial mining stage.

**Figure 3 sensors-22-05009-f003:**
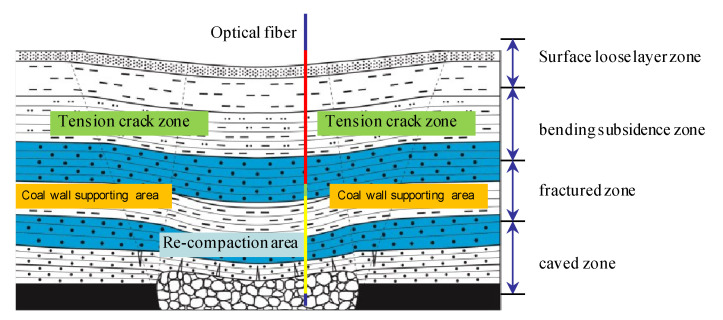
Vertical zoning division of overlying strata caused by mining.

**Figure 4 sensors-22-05009-f004:**
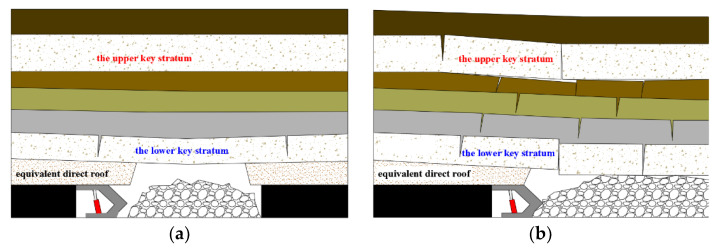
Schematic diagram of roof fracture of multi-key layer structure (**a**) initial weighting process (**b**) period weighting process.

**Figure 5 sensors-22-05009-f005:**
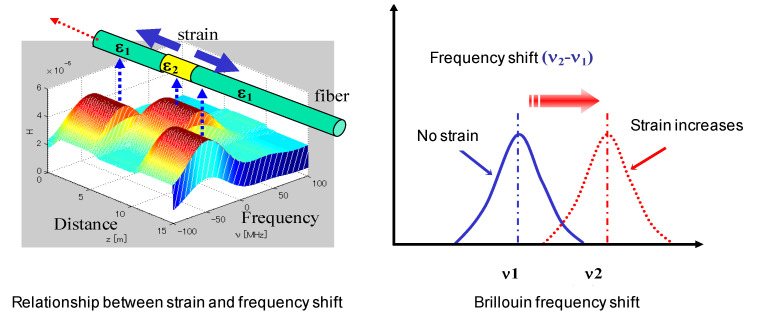
Principle of strain monitoring with BOTDA technology.

**Figure 6 sensors-22-05009-f006:**
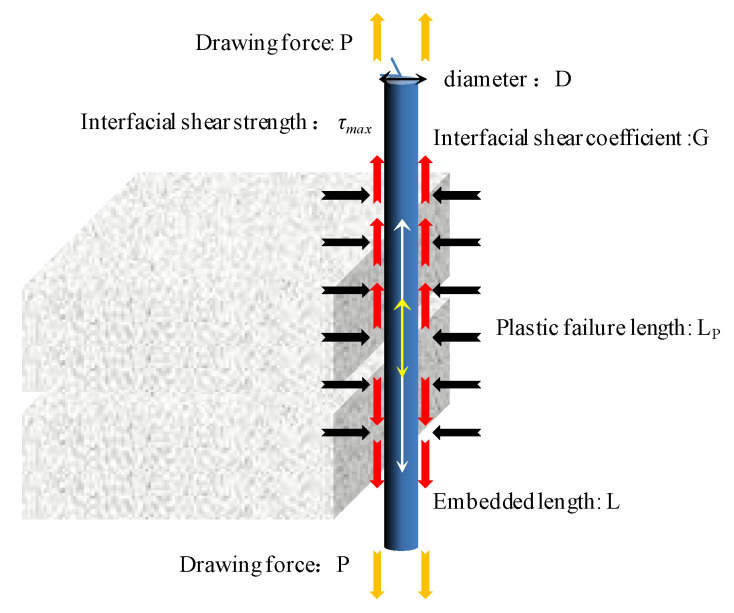
Stress analysis of optical fiber in rock mass.

**Figure 7 sensors-22-05009-f007:**
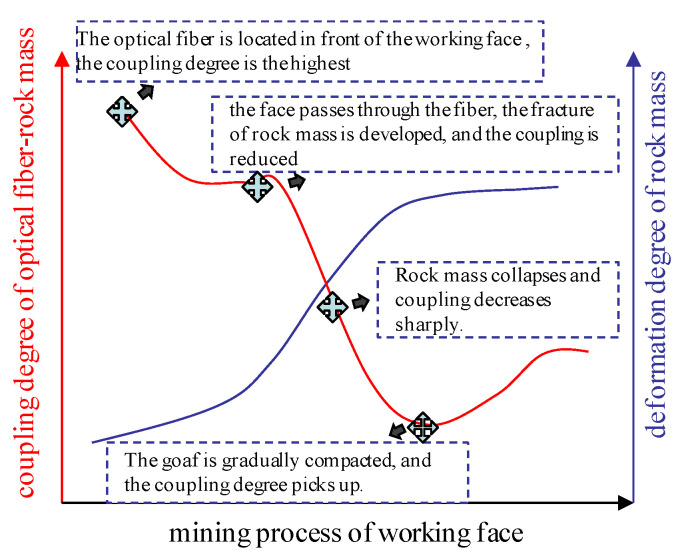
Theoretical model of coupling relationship between rock and optical fiber.

**Figure 8 sensors-22-05009-f008:**
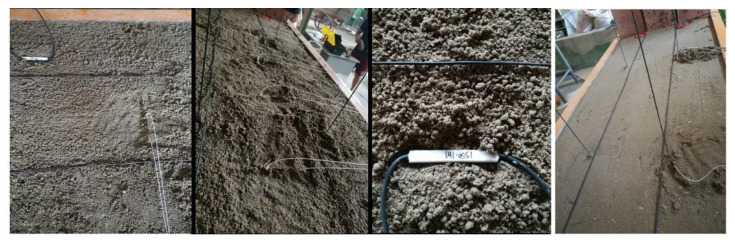
Model test system and vertical optical fiber arrangement.

**Figure 9 sensors-22-05009-f009:**
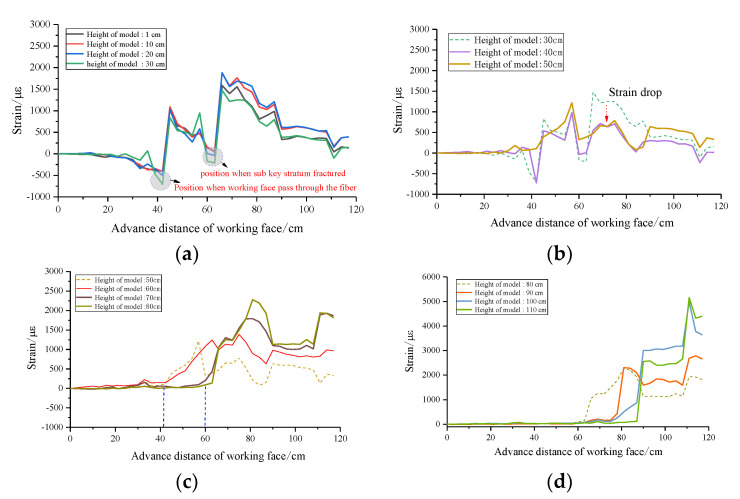
Strain distribution characteristics of rock mass detected with optical fiber at different heights. (**a**) Floor region and caving zone. (**b**) Top airspace area. (**c**) Mining-affected area under main key stratum. (**d**) Mining-affected area above main key stratum.

**Figure 10 sensors-22-05009-f010:**
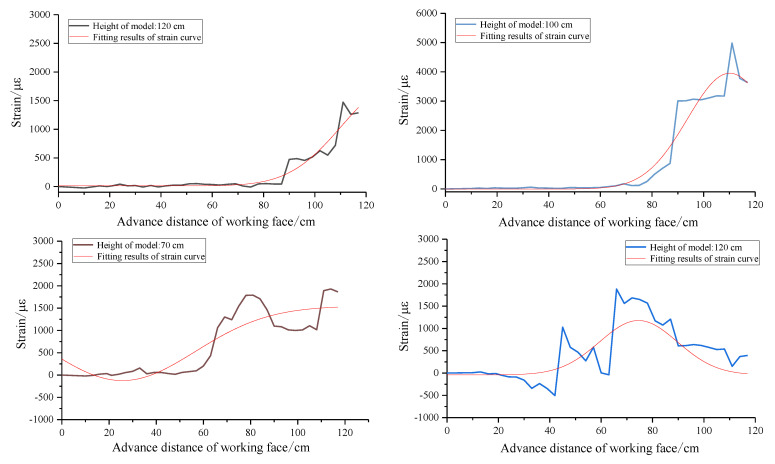
Strain fitting functions in different vertical zones.

**Figure 11 sensors-22-05009-f011:**
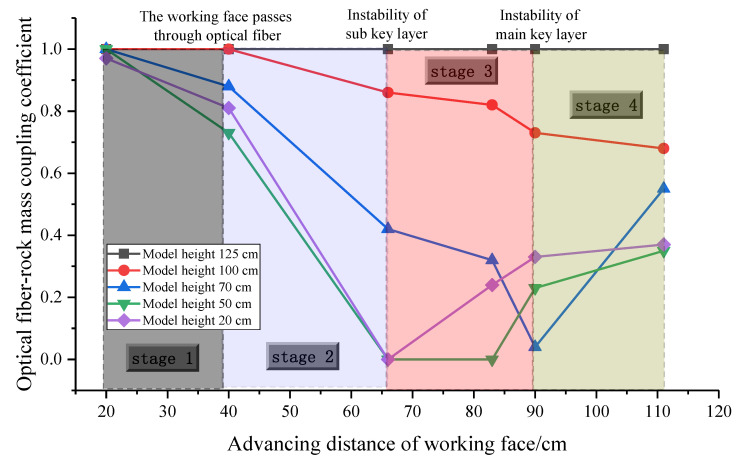
Characteristics stage division for the deformation process of overburden based on the analysis of fiber–rock coupling.

**Figure 12 sensors-22-05009-f012:**
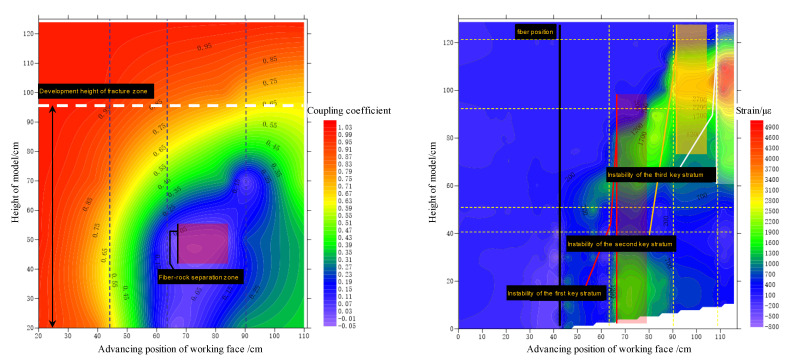
Spatiotemporal evolution of fiber–rock coupling coefficient with stope variation.

**Table 1 sensors-22-05009-t001:** Empirical equations for calculating the height of water-conducted fracture zone.

Compressive Strength/MPa	Rock Classification	Roof Management Method	Height of Water-Conducted Fracture Zone/m
40–60	limestone, sandy conglomerate, sandy shale, siliceous quartzite	Total collapse method	$H=\frac{100M}{2.4n+2.1}+11.2$
20~40	sandy shale, argillaceous sandstone, shale	Total collapse method	$H=\frac{100M}{3.3n+3.8}+5.1$
<20	weathered rock, argillaceous sandstone, quaternary loose layer	Total collapse method	$H=\frac{100M}{5.1n+5.2}+5.1$

**Table 2 sensors-22-05009-t002:** Evaluation table of coupling relationship between rock mass and optical fiber.

Coupling Coefficient/*k*	Coupling Relationship between Rock Mass and Optical Fiber 90 111
1 0.9~1	Full coupling contact Extra-strong coupling contact
0.75~0.9	Strong coupling contact
0.5~0.75	Moderate coupling contact
0.1~0.5	Weak coupling contact
0	Completely detached

**Table 3 sensors-22-05009-t003:** Distribution of fiber–rock coupling coefficient in deformation evolution.

	Advancing Position/m	21		45		66		84		90		111	
Model Height/cm		Strain /με	k_j_	Strain /με	k_j_	Strain /με	k_j_	Strain /με	k_j_	Strain /με	k_j_	Strain /με	k_j_
120		11	1	4	1	38	1	39	1	61	1	56	1
100		24	1	13	1	130	0.86	1202	0.82	3002	0.7	4921	0.68
70		19	1	62	0.8	1086	0.39	1727	0.59	1103	0.4	1902	0.72
50		−6	1	80	0.7	470	0	88	0	625	0.2	314	0.35
20		−34	0.9	−424	0.8	1884	0	1086	0.24	607	0.3	174	0.37

## Data Availability

Not applicable.

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
