# Peer review of "Study on Quantitative Characterization of Coupling Effect between Mining-Induced Coal-Rock Mass and Optical Fiber Sensing"

_sensors, 2022, doi:10.3390/s22135009_

Round 1

Reviewer 1 Report

The paper is interesting and useful to the monitoring of mine pressure, which is  well written based on the theoretical analysis and laboratory simulation tests. I suggest that the current form of the paper can be considered to be accepted before minor revisions.

(1) The use of the formulas in the paper shall indicate the sources.

(2) In Table 2, please give the explaining of the choosing values coupling coefficient and the determination of coupling relationship between rock mass and optical fiber.

(3) In Fig. 10, what kind of functions were used to fit the strain data with the advance distance of working face.

(4) For the Conclusions, the improvement is needed. The current conclusions consist of description of main work. However, the qualified conclusions should consist of the qualitative or quantitative objective results based on the work in the study.

Author Response

Question 1 reply: Formula references have been added to the paper.

Question 2 reply: Agree with the expert's proposal, and the basis for classification of coupling coefficient has been added in front of Table 2. “According to the corresponding relationship between multiple model tests phenomena and coupling coefficient, the gradation of coupling coefficient is determined as shown in Table 2. ”

Question 3 reply: Gauss function is used for fitting.

Question 4 reply: Agree with the expert's proposal, three conclusions have been revised:

(1) The coupling effect of fiber-rock can be divided into three stages. In the first stage, there is no discontinuous deformation in the surrounding rock where the optical fiber is located. In the second stage, when the surrounding rock begins to produce discon-tinuous deformation, the shear layer of the rock interface gradually occurs shear failure and interface peeling. In the third stage, when the rock mass is unstable and cracked, the interface between the optical fiber and the rock mass will slide relatively.

(2) The strain curves of different heights in vertical direction can be divided into five zones according to the variation trend. It is because of the great difference of coupling relationship between fiber and rock in the five different regions, the strain difference of optical fiber detection in different regions is caused. Area 1 is the original rock stress zone; area 2 is the bending subsidence zone; the sum of area 3 and 4 is the fractured zone; area 5 is the coal seam floor.

(3) The quantitative index of coupling between mining rock mass and fiber is proposed: coupling coefficient. The coupling coefficient of 0.65 is taken as the boundary between the fractured zone and the unbroken area of mining overburden, that is, the upper limit of the development height of the water conducted fractured zone.

Reviewer 2 Report

This manuscript deals with the problem about the coupling effect between rock and optical fiber for monitoring the rock deformation. The problem falls into the topic of the journal. However, the manuscript cannot be accepted at present form. The comments are as follow:

1 Introduction: many studies about the coupling effect between rock and optical fiber have been conducted. However, they were not presented in the literature review.

2 Compared with the previous studies on the coupling effect, the new contribution of this study should be underlined.

3 Section 2 has should be simplified.

4 Section 3.2.1, the authors claimed that ‘After calculation, it can be concluded that: the shear strength of rock-fiber interface is 7.2 327 kPa, which increases with the increase of confining pressure. When the shear stress be- 328 tween the fiber and rock mass exceeds 7.2 kPa, the fiber will slide from the rock mass.’ The shear strength does not vary for different rocks?

5 Section 3.3 model test: Many details about the model test are missing. The descriptions about the working face, model size, materials, similarity ratios should be added.

6 The conclusion need to be improved. What are the new findings from this study?

Reviewer 3 Report

The article presents a research on coupling effect between rock mass and sensing optical fibre in terms of monitoring of subsidence in coal mining industry. It presents an overview of overburden subsidence state of knowledge and the idea of sensing optic fibre application in mining industry. The key part of the article is a research on the coupling effect, based on a lab testing facility. A result of the test is a coupling coefficient that might be used in similar real-life applications. The article might be a great contribution to the subject of ground subsidence monitoring in mining industry. However, it needs some minor improvements, mostly in terms of layout and formatting:

·        Lines 80-147: This section contains some basic mechanics. Consider shortening it, as the article is already quite long.

·        Figures: Improve formatting. Figures should be also slightly bigger and some of them has to be bigger, as captions on them are barely readable (figures 3, 4, 8, 12).

·        Lines 142-147: Is this the best place for this data. It looks like it does not belong here. Also presentation of this data in a form of a table will be appreciated.

·        I think that figures should be recalled before they are placed in the text. It makes no difference to me, but please check the publisher's policy (e.g. lines 307 and 315, but not only if I remember well).

·        Lines 392-404: Is it necessary to divide into subsections if they are only one paragraph each?

·        Lines 393-397: Did you consider placing all data about the longwall in one place and presenting it in a table?

·        Lines 513-514: Please improve the formatting of the table (and other tables too).

·        Lines 564-583: Conclusions are quite short. The part about the coupling effect and coupling coefficient might be improved.

Round 2

Reviewer 2 Report

1 The use of English should be improved.

Author Response

According to experts' opinions, the English expression of the manuscript is revised.